# New Approaches to Diabetic Nephropathy from Bed to Bench

**DOI:** 10.3390/biomedicines10040876

**Published:** 2022-04-09

**Authors:** Jun-Li Tsai, Cheng-Hsu Chen, Ming-Ju Wu, Shang-Feng Tsai

**Affiliations:** 1Division of Family Medicine, Cheng Ching General Hospital, Taichung 407, Taiwan; ss881056@thu.edu.tw; 2Division of Family Medicine, Cheng Ching Rehabilitation Hospital, Taichung 407, Taiwan; 3Division of Nephrology, Department of Internal Medicine, Taichung Veterans General Hospital, Taichung 407, Taiwan; cschen920@vghtc.gov.tw (C.-H.C.); wmj530@vghtc.gov.tw (M.-J.W.); 4Department of Life Science, Tunghai University, Taichung 407, Taiwan; 5Department of Post-Baccalaureate Medicine, College of Medicine, National Chung Hsing University, Taichung 402, Taiwan; 6School of Medicine, National Yang-Ming University, Taipei 112, Taiwan

**Keywords:** chronic kidney disease, diabetic nephropathy, hypoxia, anemia, hypoxia, hypoxia-inducible factor (HIF), glomerular hyperfiltration

## Abstract

Diabetic nephropathy (DN) is the main cause of end-stage kidney disease (ESKD). DN-related ESKD has the worst prognosis for survival compared with other causes. Due to the complex mechanisms of DN and the heterogeneous presentations, unmet needs exist for the renal outcome of diabetes mellitus. Clinical evidence for treating DN is rather solid. For example, the first Kidney Disease: Improving Global Outcomes (KDIGO) guideline was published in October 2020: KDIGO Clinical Practice Guideline for Diabetes Management in Chronic Kidney Disease. In December of 2020, the International Society of Nephrology published 60 (+1) breakthrough discoveries in nephrology. Among these breakthroughs, four important ones after 1980 were recognized, including glomerular hyperfiltration theory, renal protection by renin-angiotensin system inhibition, hypoxia-inducible factor, and sodium-glucose cotransporter 2 inhibitors. Here, we present a review on the pivotal and new mechanisms of DN from the implications of clinical studies and medications.

## 1. Introduction

Diabetes mellitus (DM) is the leading acquired risk factor for accelerated progression of chronic kidney disease (CKD). In the United States, DM accounts for 30% to 50% of end-stage kidney disease (ESKD) cases [1]. Patients receiving dialysis due to DM have worse survival compared with ESKD due to non-DM related causes. According to a recent study, DM with early kidney involvement shortens life expectancy by 16 years [1]. According to a National Health and Nutrition Examination Survey, United States population-based study, the progression of DM-related nephropathy is associated with higher mortality [2]. Among individuals with DM but not kidney disease, standardized mortality is up to 11.5%, compared with the control group (people without DM or kidney disease, 10-year cumulative all-cause mortality is 7.7%) [2]. Among individuals with both diabetes and kidney disease, standardized mortality reaches 31.1% with an absolute risk difference to the control group of 23.4% [2]. Therefore, prevention or treatment of diabetic nephropathy (DN) is mandatory for patients with DM.

The poor renal outcomes of DM has not improved over decades [3]. According to the American National Health Interview Survey [3], the proportions of all five complications (lower-extremity amputation, ESKD, acute myocardial infarction, stroke, and death from hyperglycemia crisis) has declined from 1990 to 2010. But the decline in ESKD remains small, and even increases among the elderly. Despite advanced clinical care, improvements in the health care system, and greater efforts in health promotion [4,5,6], ESKD cases still keep increasing. Specifically, when expressed in terms of the absolute number of cases, the annual ESKD cases grew by 32,434 between 1990 and 2010 in the United States [3]. In an analysis in which rates are expressed per 10,000 persons in the overall population, the rate of ESKD have increased by 90.9% (from 1.1 to 2.1 cases per 10,000 population) [6]. Clearly unmet needs exist for clinical care of renal outcomes for DM patients. The statistics also reflect the complex and unknown mechanisms underlying DM-related chronic kidney disease (CKD) [7]. In this review, we focused on some pivotal and new mechanisms of DN, including glomerular hyperfiltration theory, renal protection by renin-angiotensin system inhibition (RASi), hypoxia-inducible factor (HIF), and sodium-glucose cotransporter 2 inhibitors (SGLT2i). The above four mechanisms/medications have been listed as the four breakthrough discoveries in nephrology after 1980 [8]. 

In the guidelines from National Kidney Foundation in 2007, a new terminology to describe kidney disease attributable to DM (diabetic kidney disease, DKD) has been introduced to replace diabetic nephropathy (DN) [9,10]. DKD is a diagnosis based on clinical and laboratory findings (glomerular filtration rate (GFR) and albuminuria) [11] which, along with clinical characteristics of diabetes (such as diabetes duration and the presence of diabetic retinopathy), increases the likelihood of kidney involvement [11]. DN is the presence of a single, well-defined, identifiable kidney disease identified by progressive glomerular nephropathy directly related to diabetes [12]. DN should only be used when a patient has a biopsy confirmed nephropathy and should be accompanied by Tervaert’s classification [13]. In our present review, we preferred DN over DKD to specify the DM-related CKD.

## 2. From Bed to Bench: Implications from Clinical Studies

In the past 20 years, very few clinical studies have focused on the renal outcomes of DM or DM control in patients with advanced CKD or ESKD. That is why the ‘Kidney Disease: Improving Global Outcomes (KDIGO) Clinical Practice Guideline for Diabetes Management in Chronic Kidney Disease’ represents the first KDIGO guideline published in Oct, 2020 [14]. DN outcomes have changed in recent years at a pivotal time due to new therapies [11,15]. First, dipeptidyl peptidase-4 Inhibitors (DPP4i) lower albuminuria but without GFR benefits [16,17]. SGLT2is have reported better pre-specified renal outcomes, including albuminuria and GFR, from studies like the cardiovascular outcome trials (CVOTs) (empagliflozin for EMPA-REG [18], canagliflozin for CANVAS [19], and dapapagliflozin for DECLARE-TIMI 58 [20]). In addition, better renal outcomes have been demonstrated by treatment with SGLT2is (CREDENCE [21], and DAPA-CKD [22]), and third generation nonsteroidal selective mineralocorticoid receptor antagonists (MRA) (finerenone from FIDELIO-DKD [23] and FIGARO-DKD [24]). The renal outcome study from the EMPA-KIDNEY study will be published later this year [22]. Recently, the EMPA-KIDNEY study stopped early due to evidence of efficacy [25]. The SGLT2is are the only Class 1A recommendation in the KDIGO guideline, whereas RASi is listed as Class IB [14]. Atrasentan (an endothelin A receptor antagonist) reduces the risk of renal events (doubling of serum creatinine) in patients with DM and CKD as found in the SONAR trial [26]. Glucagon-like peptide-1 receptor agonists (GLP-1RA) (LEADER [27], ELIXA [28], SUSTAIN-6 [29], REWIND [30], and EXSCEL [31]) also showed renal benefits in the CVOTs [32]. In patients with type 2 DM and moderate-to-severe CKD, dulaglutide reduced decline in GFR compared to basal insulin (AWARD-7 study) [33]. A renal outcome trial (FLOW study) from Semaglutide (a new GLP-1RA) will be published in 2024. Therefore, now is a good time to review DN because of the many clinical studies (DPP4i, SGLT2i and GLP-1RA) focusing on this issue with benefits on renal outcomes. All pivotal and new mechanisms of DN are summarized in Figure 1.

## 3. A Paradigm Shift in Nephrology: Glomerular Hyperfiltration

### 3.1. Glomerular Hyperfiltration in CKD

In 1996, Barry M. Brenner put forward a concept, entitled “The hyperfiltration theory: A paradigm shift in nephrology” [34]. He proposed that glomerular hyperfiltration aggravates the progression of renal damage in most CKD, especially in DN [34]. CKD is defined as the presence of kidney damages for three or more months based on all renal insults [35]. That means reduced nephron numbers in CKD accompanied by maladaptive glomerular hemodynamic hyperfiltration [36]. This vicious cycle further threatens residual renal function. 

Compensatory increased GFR is due to changes in the glomerular capillary plasma flow rate and mean glomerular capillary hydraulic pressure [34]. It is also due to adaptive afferent arteriolar vasodilation (vascular factors, including nitric oxide bioavailability, COX-2 prostanoid, kallikrein-kinin, atrial natriuretic peptide, angiotensin (1–7), hyperinsulinemia and tubular signals, including the inhibition of tubuloglomerular feedback (TGF)) and efferent arteriolar vasoconstriction (vascular factors, including angiotensin-II, thromboxane A2, endothelin-1, and reactive oxygen species) [37]. Subsequent overwork of remnant nephrons experience greater glomerular hyperfiltration, which further threatens the residual renal function. This vicious cycle needs to be stopped to prevent CKD progression.

### 3.2. Glomerular Hyperfiltration in DM Related CKD

Glomerular hyperfiltration is a typical and very early finding of DN [37]. Around a 25–50% elevation of GFR is noticed early in patients with type 1 DM [38]. It is defined as filtration levels between 120 and 140 mL/min/1.73 m^2^ [37]. Its progression triggers glomerular sclerosis [39]. Once microalbuminuria starts (traditional definition of DN—the incipient stage of DN), about half of the nephrons will have already lost. Therefore, how to stop glomerular hyperfiltration has been a hot topic since the breakthrough concept was put forward in 1996 [40]. Causes of glomerular hyperfiltration in DM include elevated RAS [39], imbalance between afferent and efferent arterioles resistance [37], and impaired TGF [41]. Insulin-like growth factor 1 [42], sorbitol [43], and advanced glycation end products [44] are also associated with glomerular hyperfiltration in DN. Recently, impaired TGF has been a target for new medications for DM, including SGLT2is and GLP1-RAs. The main physiological task of kidneys is to reabsorb as much glucose as possible so that the normal urine is glucose-free [45]. The glucose is non-protein-binding, and non-complexed with macromolecules filtered freely at glomeruli. Therefore, in patients with DM with glycosuria, the high urinary glucose should be reabsorbed by SGLT-1 (10%) and SGLT-2 (90%) in the proximal tubule. However, the enhanced tubular sodium reabsorption causes extracellular fluid volume expansion and then further leads to glomerular hyperfiltration [38,46]. The decreased sodium delivery to distal tubule by activating the TGF mechanism in the macula densa could also raise the GFR via vasodilatation of the afferent arteriole [47]. Recently, the renal benefits of SGLT2i and GLP1-RA have been demonstrated, based on restoring TGF and then stopping glomerular hyperfiltration. 

### 3.3. Treatment of Glomerular Hyperfiltration

Treatments reducing glomerular hyperfiltration have been used to stop or prevent DN since 1980s. These include old and new therapies such as: aggressive control of blood pressure, low salt diet [48], low animal protein diet [48,49], the usage of RASi [50,51,52,53,54,55], SGLT2i [56], atrasentan (ET-1 antagonist) [26], finerenone [23,24], and GLP-1RA [57]. First, the breakthrough concept of glomerular hyperfiltration is central to the use of RASi to prevent and treat DN (through efferent arteriole vasodilation) as reported by Taguma et al. in 1985 [50]. The hallmark study was published in 1993, which showed that captopril protected against deterioration in renal function in patients with insulin-dependent diabetic nephropathy, and was significantly more effective than blood-pressure control alone [51]. Later studies like IDNT [52], RENAAL [53], IRMA-2 [54], and BENEDICT [55] all supported the renal benefit of RASis for DN patients. As a result, RASis have remained the typical and basic treatment of DN over the past 20 years. To obtain better long-term renal outcomes, clinicians should tolerate the initial reduction of GFR up to 30% [14]. 

In addition to RASis, other emerging pharmacological treatments have also focused on glomerular hyperfiltration. For example, SGLT-2is reduce hyperfiltration in DN by restoring TGF through afferent arteriole vasoconstriction [56]. SGLT2is inhibit sodium uptake (through SGLT2, SGLT1, and Na+/H+ exchanger isoform 3 (NHE3) transports [58]) in the proximal tubule, which leads to the prevention of serum sodium related hyperfiltration. It reduces intraglomerular pressure by 19% [59,60]. Another example is the long-acting GLP-1RA which has reported renal protection through the inhibition of NHE3 [61], which in turn reduces intraglomerular hyperfiltration. Finally, endothelin-A receptor antagonists from the SONAR study showed reduced risk of renal events in patients with DN [26]. These antagonists have vasoconstrictive effects on the efferent arterioles leading to less glomerular hyperfiltration. All the above medications used to stop glomerular hyperfiltration have an initial drop in GFR and better long-term renal outcomes. 

## 4. Anemia in DN: Implications from HIF Stabilizer and SGLT2i

### 4.1. Anemia in CKD and in Particular in DM-Related CKD

Anemia is a common and major complication for CKD patients. It progresses as renal function deteriorates. Such anemia starts in CKD stage 3 and prevalence rises to 67% in stage 5 [62]. In some countries [63], the prevalence of CKD-related anemia is as high as 79.2% in stage 4, and 90.2% in stage 5. DM is a leading cause of ESKD, but no renal anemia guidelines have focused on this population (DM related CKD). Recently, a number of studies have reported that SGLT2is have renal benefits in addition to sugar control, including EMPA-REG [18], DECLARE-TIMI 58 [20], CANVAS [19], and CREDENCE [21]. The renal benefits of SGLT2is can be linked to the alleviation of renal anemia [64].

Renal anemia may develop earlier and be worse in DM-related CKD compared with non-DM-related CKD [58,65]. As reported in a cross-sectional study on DM patients by Thomas et al. up to 23% of patients have anemia [66]. Untreated CKD related anemia is associated with increased mortality and morbidity and patients with DN related anemia are at particularly increased risk [67]. The mechanism of anemia due to DN is summarized in Figure 2. Generally, all risk factors for renal anemia in the general population may also cause anemia in DN patients. These risk factors include aging kidneys, source deficiency for RBC production, and blood loss (particularly bleeding tendencies due to uremic coagulopathy, or antiplatelet related coagulopathy [68], and advanced glycation end production (AGE) related RBC deformability [69]). Impaired O_2_ sensing in EPO production may be due to the following: diabetic autonomic neuropathy [70], reduced stabilization of HIF-1 [71], the adverse effects of medications of RASi [72], and reactive oxidative stress (ROS) related EPO insufficiency [65]. EPO related factors are a major causes of anemia. Such factors include urinary EPO excretion [66], inflammatory cytokine (e.g., interleukin-1, tumor-necrosis factor, and interferon-γ) [73,74], and AGE related EPO resistance [75]. Based on the above issues, the onset of anemia in DN has been reported to occur sooner when compared with non-DM related CKD [65,72,76,77,78]. Moreover, anemia prevalence is higher in stage 3 CKD when compared with non-DM related CKD [63], 53.5% vs. 33.1% (*p* = 0.001), respectively.

### 4.2. The Impact of Renal Anemia on Renal Function

There is no consensus about the direct impact of renal anemia on renal function deterioration. This controversial issue is due to the multiple and complex causes of CKD, small cases numbers, short duration of follow-up, and inconsistent outcome setting [79,80,81,82,83]. In a well-defined study (only from DN-related anemia) [84], it was demonstrated that anemia due to DN is an independent predictor for progression to ESKD. When analyzed by Cox proportional hazards models [84], the baseline hemoglobin concentration was correlated with the subsequent development of ESKD, and the adjusted hazard ratio was 0.90 (95% CI 0.84–0.96, *p* = 0.0013). Similarly, from other studies [75,85,86], renal anemia was shown to cold contribute to the progression of renal dysfunction in DM-related CKD. In a prospective cohort study, low EPO levels could also predict faster renal function decline independently of established prognostic factors including GFR, albuminuria, and hemoglobin in DN with anemia [87]. Kuriyama et al. found that reversal of anemia by EPO can slow the progression of CKD [79]. One observational and one randomized controlled study both have identified that EPO treatment slows the onset of dialysis in DN-related anemia [82,88]. In 2016, in a study on DN animal models, EPO was reported to have suppressed the inflammatory response, along with oxidative damage in an animal model [89]. An EPO receptor activator when applied continuously reduces tubulointerstitial fibrosis in a DN animal model (*db/db* mouse) [90]. Therefore, the renal anemia in DN is associated with renal deterioration with a possible causal effect and the treatment of DN related anemia with EPO may slow the deterioration of DN. 

### 4.3. Effect of SGLT2is on Renal Anemia in DN

In the CVOTs of the SGLT2is, all four SGLT2is have produced a modest increase in hematocrit (2–4%) [64]. This effect cannot fully be explained by the initial diuretic effect related to hemoconcentration. The EPO level increased after the initiation of dapagliflozin, reaching a plateau in 2 to 4 weeks [91]. A gradual increase in hemoglobin beyond week 4 indicated an EPO effect of SGLT2i. In another SGLT2i trial for heart failure (DAPA-HF) [92], dapagliflozin corrected anemia more often than placebo group and it improved heart outcomes, irrespective of anemia status at baseline. In a pooled study of 5325 patients from 14 placebo-controlled trials [93], dapagliflozin corrected anemia in 52% of patients with anemia at baseline (placebo: 26%). In a systematic review and meta-analysis [94], each SGLT2i (including canagliflozin, dapagliflozin, empagliflozin, and ipragliflozin) led to a significant increase in the hematocrit level when compared to placebo (MD 1.32%, 95% CI = 1.21–1.44, *p* < 0.00001, considerable heterogeneity-I^2^ = 99%). Therefore, SGLT2is can relieve renal anemia with solid evidence, which can reduce the progression of DN. 

### 4.4. Two New Treatments for Renal Anemia (HIF Stabilizer and SGLT2i)

Two hallmarks of renal anemia were EPO deficiency and dysregulation of iron [95]. In current clinical practice, the optimal treatment algorithms for renal anemia are based on EPO administration and iron supplementation. However, there are many concerns about current EPO and iron therapy for renal anemia. First, patients experienced greater risk for death, adverse cardiovascular effects, and stroke when higher dose of EPO was used to target a hemoglobin level of great than 11 g/dL [96]. Second, there are also some disadvantages regarding cardiac complications and infections that result from iron over supplementation [97]. The culprit of dysregulation of iron the reduced excretion of hepcidin in CKD [98]. The increased hepcidin reduces iron absorption in the duodenum and releases iron from the macrophages, which interact with and inactivate the iron export protein ferroportin [99,100]. The cause of dysregulation of iron in CKD is due to high hepcidin without successful treatment of this problem until the administration of a HIF stabilizer. 

Research from early in 2008 demonstrated that patients with ESKD living at high altitude either increase endogenous EPO production or respond better to EPO administration [101]. Altitude-induced hypoxia reduces EPO requirements in ESKD patients with treatment-refractory anemia [102]. In addition, Tibetan people with a natural prolyl hydroxylase domain (PHD2) mutation had higher blood HIF, which led to more EPO and higher blood hemoglobin [103,104]. This congenital mutation causes the Tibetan population to adapt to the chronic hypoxia of high altitude. Based on these epidemiological and genetic studies, high altitude that induces higher blood HIF is of benefit for renal anemia in patients with ESKD. Roxadustat, a small molecule HIF PHD inhibitor, is a medication to treat renal anemia. It can reversibly bind to and inhibit HIF-PHD enzymes that are responsible for the degradation of transcription factors in the HIF family under normal oxygen conditions [105]. Roxadustat can treat renal anemia in both dialysis-dependent CKD [106] and non-dialysis dependent CKD [107,108]. At least, the effect of Roxadustat on renal anemia is like EPO. Interestingly, the HIF stabilizer can target two hallmarks of renal anemia simultaneously, including increasing blood EPO levels and reducing hepcidin. From a meta-analysis and systemic review [109], roxadustat can significantly reduce hepcidin levels (−31.96 ng/mL, 95% CI (−35.05 ng/mL, −28.87 ng/mL), *p* < 0.00001) and ferritin (−44.82 ng/mL, 95% CI (−64.42 ng/mL, −25.23 ng/mL), *p* < 0.00001). This is the first time we have increased hepcidin in clinical practice. In a preclinical study [110], Enarodustat (PHD inhibitor) can activate HIF and then protect against metabolic disorders and associated kidney disease in obese, type 2 diabetic mice. 

In a recent review [111], DM was found to cause hypoxia in the renal cortex, increasing oxidative stress leading to an imbalance between HIF-1α/HIF-2α. Reduced HIF-2α or increased HIF-1α further causes a reduction in EPO and dysregulation of the ferrokinetics. SGLT2is reverse the imbalance between HIF-1α/HIF-2α, consequently producing more EPO and improving iron usage. The renal benefits of SGLT2is can be partially explained by the relief of renal anemia, via balancing the HIF-1α/HIF-2α ratio. The overexpression of HIF-1α was considered as an inflammatory effect on renal injury and the activation of HIF-2α can counteract the inflammation and reduce injury in CKD [112,113]. In a preclinical study [114], an SGLT2 inhibitor (luseogliflozin) was shown to reduce the protein expression of HIF-1α expression in the human renal proximal tubular epithelial cells. Additionally, luseogliflozin also inhibited HIF-1α gene expression, including PAI-1, VEGF, GLUT1, HK2, and PKM. It also attenuated cortical tubular HIF-1α expression in *db/db* mice [114]. Moreover, SGLT2is can upregulate both SIRT1 and AMPK [115,116]. Increasing SIRT1 and AMPK can further suppress HIF-1α and promote HIF-2α [116,117,118].

## 5. Energy Demand–Generation Imbalance, Hypoxia, and Reactive Oxidative Stress

### 5.1. Mitochondria Dysfunction and Increased Energy Wasting in DN

Mitochondria are responsible for more than 90% of energy production, by oxidative phosphorylation, in the human body. However, mitochondrial DNA is susceptible to ROS related damage. Mitochondrial dysfunction is considered a contributing factor in many diseases, including renal disease [119] and DM [120]. Mitochondria are most abundant in the kidney, second only to the heart [121]. In the kidney, both the mitochondrial volume density [122] and Na+/K+ ATPase activity [123] are located mostly in the proximal and distal tubules. Therefore, it is reasonable that mitochondrial dysfunction causes decreased ATP production, alterations in cellular function, and the loss of renal function [124]. Risk factors of mitochondria dysfunction and energy wasting in DN are as follows [125,126]: First, high plasma glucose can directly damage renal tubular cells and then cause metabolic and cellular dysfunction [127]. Second, the overproduction of ROS are interlinked mechanisms that play important roles in the progression of DN. Hyperglycemia causes the overproduction of electron donors (nicotinamide-adenine-dinucleotide and flavin-adenine-dinucleotide) by the Krebs’ cycle, and this condition surpasses the capacity of the mitochondrial electron transport chain [128]. Subsequently, the excess electrons are transferred to oxygen, followed by increased ROS. Third, in DN, to maintain glucose-free urine, more energy is needed for sodium reabsorption in the proximal and distal tubules through mitochondrial oxidative metabolism [129]. To achieve this goal, SGLT2 and SGLT1 are over-expressed for glucose and sodium reabsorption [130], resulting in energy wasting in DN. Finally, as renal function deteriorates (accompanied by fewer viable nephrons and less energy production), the energy imbalance also worsens. Mitochondrial biogenesis declines following the progression of DN [131]. Therefore, administering compounds [132,133,134] that stimulate mitochondrial biogenesis can restore mitochondrial and renal function in DN, including AICAR (an AMPK activator) in *db/db* diabetic mice and *ob/ob* obese mice [135]. SGLT2is are thought to balance sodium and calcium homeostasis [136] and rescue mitochondrial function. SGLT2is may also reduce ATP consumption for less energy wasting by inhibiting sodium reabsorption and metabolic stress in the proximal tubular epithelial cells [129].

### 5.2. Renal Hypoxia and HIF in DN

Hyperglycemia results in increased oxygen consumption and decreased oxygen tension in the kidney. Diabetic rats displayed tissue hypoxia throughout the kidney, glomerular hyperfiltration, increased oxygen consumption, increased total mitochondrial leak respiration, and decreased tubular sodium transport efficiency [137]. Renal hypoxia is also a typical finding for the development of DN [138,139]. Renal blood flow accounts for 25% of cardiac output, maintaining enough renal blood flow and oxygen delivery to the kidneys. In other words, kidneys are very susceptible to mismatch between oxygen demand and supply, which can cause progression of DN [140]. 

According to a preclinical study, renal hypoxia can be identified much earlier than pathological changes of DN [141]. Renal hypoxia precedes albuminuria in type 1 diabetic mice [141]. In the early stage of DN, hypoxia triggers HIF-1α production, followed by increased inflammatory and fibrotic molecules [142], including TGFβ and TNF. In addition to renal hypoxia, DM is notorious for AGE-related, increased ROS [143]. The increase in ROS also triggers HIF-1α production. In a recent review [144], the role of renal hypoxia and ROS in the pathogenesis of DN was reported and this condition can be reversed by SGLT2i. SGLT2i blocks HIF-1α related hypoxia and ROS associated with renal fibrosis. 

Blood oxygenation level dependent magnetic resonance imaging (BOLD-MRI) was used to study renal hypoxia in human study since 1990s [145]. This measurement assessed a lower PO2 in the renal cortex in DN compared to healthy population [146]. R2* ratio between the medulla and cortex increased in the early stage of diabetes and decreased along with the progression of DN [146]. Additionally, the effect of SGLT2is on renal hypoxia can be detected in mathematical modeling [129,146]. Bessho et al. found that luseogliflozin attenuated cortical tubular HIF-1α expression, tubular injury, and interstitial fibronectin in *db/db* mice [114]. Luseogliflozin also inhibits hypoxia-induced HIF-1α accumulation by inhibiting mitochondrial oxygen consumption [114]. The SGLT2 inhibitors may protect DNs by targeting the relief of renal hypoxia related injury. 

## 6. Proinflammatory and Profibrotic Pathways: Interplay between HIF-1α and HIF-2α

The usage of SGLT2i or HIF stabilizers may be associated with suppressed oxidative stress and proinflammatory/profibrotic pathways. The result is amelioration of cardiac, vascular, and renal diseases [121]. In a pre-clinical study [147], stable expression of HIF-1α significantly upregulated α-smooth muscle actin expression, and reduced the E-cadherin expression in HK-2 cells during ischemia/reperfusion injury. It then induced epithelial–mesenchymal transition in renal tubular epithelial cells, followed by renal fibrosis. In another study of murine models of ischemia/reperfusion injury and unilateral ureteral obstruction [148], the increased expression of HIF-1α in tubular epithelial cells was associated with selective shedding of microRNA-23a (miRNA-23a)-enriched exosomes in vivo and systemic inhibition of miRNA-23a prior to ischemia/reperfusion injury attenuated by tubulointerstitial inflammation. The HIF-1α-dependent release of miRNA-23a-enriched exosomes from hypoxic tubular epithelial cells stimulates macrophages to promote tubulointerstitial inflammation. In another animal model of lupus nephritis [149], HIF-1α promoted mesangial cell growth via the induction of proliferation and inhibition of apoptosis, and then played an important role in the pathogenesis. 

As for DN, in an animal model of type 1 DN [150], OVE26, HIF-1α inhibitor, attenuated kidney glomerular hypertrophy, mesangial matrix expansion, extracellular matrix accumulation, and urinary albumin excretion. This study suggested that pharmacological inhibition of HIF-1α could improve the clinical manifestation of DN [150]. In an animal model of type 2 DN (*db/db* mice) [151], the small GTPase Rho and its effector Rho-kinase was activated under hypoxic conditions, and caused diabetic glomerulosclerosis via HIF-1α accumulation. Taken together, persistent high HIF-1α will activate inflammation-related cytokines, profibrotic gene transcription, macrophage infiltration, and collagen deposition, mesangial cell proliferation, and tubulointerstitial inflammation in DN. In addition to HIF-1α, HIF-2α is also important for renal injury. HIF-1α is ubiquitously expressed in the human body, and HIF-2α is detected mostly in vascular endothelial cells during embryonic development [152]. HIF-2α mRNA has also been detected in kidney fibroblasts, liver hepatocytes, and epithelial cells of the intestinal lumen [153,154]. HIF-2α, rather than HIF-1α, is associated with erythropoietin [155,156]. In spontaneously hypertensive rats [157], metformin reduced proteinuria in vivo and increased VEGF-A production in rat kidneys and cultured rat podocytes, probably by activating the HIF-2α-VEGF signaling pathway. Upregulated HIF-1α and downregulated HIF-2α are associated with the progression of DN [121]. The imbalance between HIF-1α and HIF-2α is summarized in Figure 3. 

Potential treatment-associated HIF-related proinflammatory and profibrotic pathways were studies recently. Cobalt inhibited HIF degeneration [158] and relieved renal injury in an obese, hypertensive type 2 diabetes rat model [159]. In an animal model, FG-4592 (Roxadustat) markedly ameliorated cisplatin-induced kidney injury as evidenced by the improvement in renal function and kidney morphology [160]. Pretreatment with roxadustat also attenuated folic acid-induced kidney damage by antiferroptosis through the Akt/GSK-3 β/Nrf2 pathway [161]. In addition, roxadustat protected against the renal ischemia/reperfusion injury by suppressing inflammation [162]. Therefore, the HIF-related pathway is an emerging target mechanism for DN in the future. SGLT2is can normalized the interplay between HIF-1α and HIF-2α (inhibitor HIF-1α and stimulate HIF-2α) to stop the progression of DN [114,117,156]. 

## 7. Conclusions

There are still unmet needs for the treatment of DN. Since 1980, four breakthrough discoveries have been made in the field of nephrology [8]: glomerular hyperfiltration theory, RASi, HIF, and SGLT2i. Pivotal and novel mechanisms are emerging for DN, including mechanisms of glomerular hyperfiltration, renal anemia, hypoxia, and energy imbalance. In addition to RASi, HIF stabilizer, and SGLT2i are believed to be paradigm shifts in DN treatment and prevention.

## Figures and Tables

**Figure 1 biomedicines-10-00876-f001:**
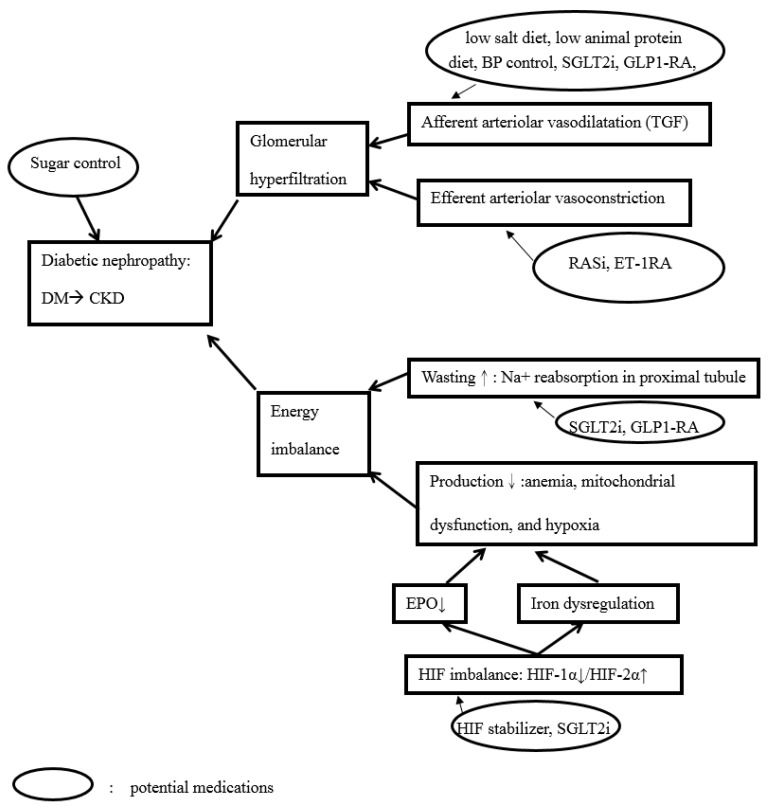
Pivotal mechanisms and potential treatments of DM related CKD. Sugar control is beneficial for DN control. Glomerular hyperfiltration (due to afferent arteriolar vasodilatation or efferent arteriolar vasoconstriction) and energy imbalance are both associated with DN. Low salt diet, low animal protein diet, controlled BP, SGLT2i and GLP1-RA are associated with afferent arteriole. On the contrary, RASi and ET-1RA are associated with efferent arteriole. The energy imbalance is due to energy wasting or too little energy production. The energy wasting can be due to excessive sodium reabsorption, and it can be stopped by SGLT2i and GLP1-RA. Low energy production can be due to anemia, mitochondrial dysfunction, and hypoxia. Renal anemia is both due to EPO deficiency and iron dysregulation. Renal anemia can be ameliorated by HIF stabilizers or SGLT2is. Abbreviations: DM, diabetes mellitus; CKD, chronic kidney disease; BP, blood pressure; SGLT2i, sodium-glucose cotransporter 2 inhibitor; GLP1-RA, GLP-1 receptor agonist; TGF, tubuloglomerular feedback; RASi, renin-angiotensin system inhibitor; ET-1RA, endothelin-1 receptor antagonist; EPO, erythropoietin; HIF, hypoxia-inducible factor. (↑: increased; ↓: decreased).

**Figure 2 biomedicines-10-00876-f002:**
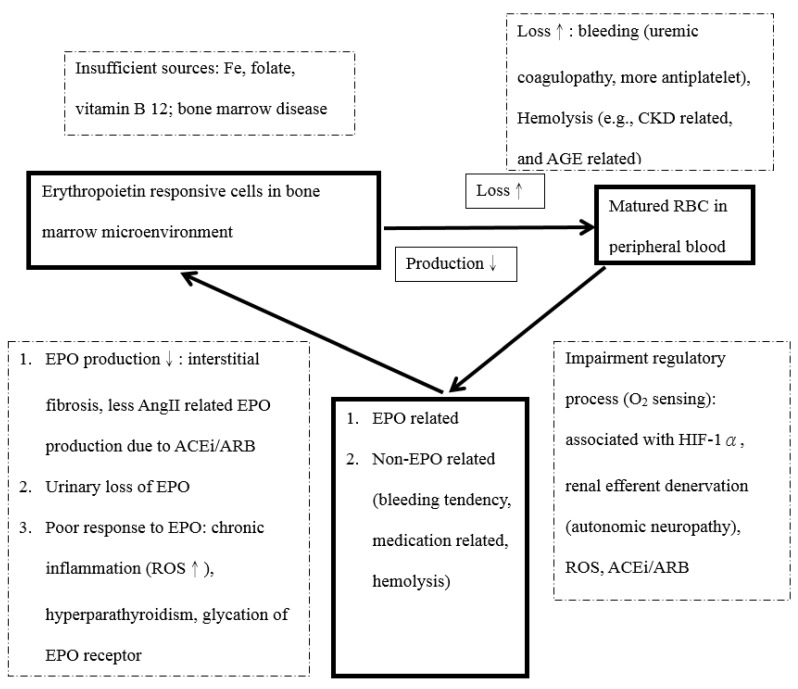
Overview for mechanisms of anemia in DM-related CKD. Patients with anemia due to either blood loss (bleeding or hemolysis) or too little RBC production (insufficient iron, folate, vitamin B12, or bone marrow disease). In addition, impaired O2 sensing can also cause renal anemia, associated with HIF-1α, renal efferent denervation, ROS, and ACEi/ARB. Moreover, reduced EPO production, increased urinary loss of EPO or poor response to EPO can also cause renal anemia. Abbreviations: EPO, erythropoietin; AngII, angiotensin II; ACEi/ARB, angiotensin converting enzyme inhibitor/angiotensinⅡreceptor blocker; ROS, reactive oxidative stress; CKD, chronic kidney disease; AGE, advanced glycation end product; RBC, red blood cell; HIF, hypoxia-inducible factor.

**Figure 3 biomedicines-10-00876-f003:**
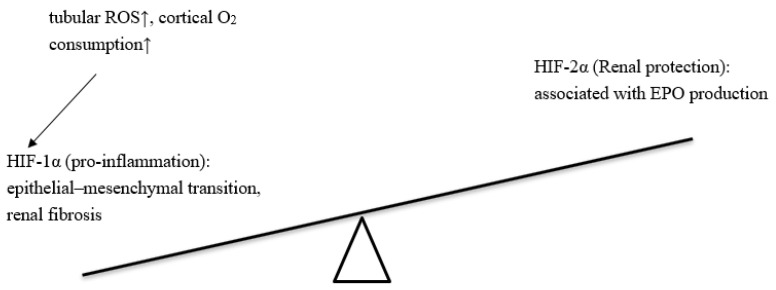
Increased HIF-1α and reduced HIF-2α in DM related CKD. Renal injury due to increased HIF-1α and reduced HIF-2α. Increased HIF-1α can be due to increased tubular ROS and increased cortical oxygen consumption. The overproduction of HIF-1α caused epithelial-mesenchymal transition and renal fibrosis. Compared to HIF-1α, HIF-2α is associated with EPO production. Abbreviations: ROS, reactive oxidative stress; HIF, hypoxia-inducible factor; EPO, erythropoietin.

## Data Availability

Not available.

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
