# Peer review of "New Approaches to Diabetic Nephropathy from Bed to Bench"

_biomedicines, 2022, doi:10.3390/biomedicines10040876_

Round 1
Reviewer 1 Report
The authors described advances in diabetic nephropathy medications and clinical and basic evidences, including the latest treatments such as SGLT2 inhibitors and HIF stabilizers. Reviewer has several concerns that need to be addressed.
1.SGLT2 has been reported to improve anemia. The authors described that correction of HIF1a/HIF2a imbalance improves anemia, but reviewer does not know the mechanism by which SGLT2 affects HIF2a. Please add an explanation and the relevant literature.
2. Protective effects of HIF-PH inhibitors on animal models of diabetic nephropathy have also been reported, suggesting a suppressive effect of HIF1 on MCP1 (e.g. PMID: 31996409). Please cite these references and discuss the significance of HIF and HIF-PH inhibitors in diabetic kidney injury.
3. There are errors in the description; second line from the bottom on page 6. “oxidative damageiananimal[87].” In figure 2, “more antiplatlet” (upper right) and “Ppoor response to EPO” (lower left).
Author Response
Reviewer 1
The authors described advances in diabetic nephropathy medications and clinical and basic evidences, including the latest treatments such as SGLT2 inhibitors and HIF stabilizers. Reviewer has several concerns that need to be addressed.
- SGLT2 has been reported to improve anemia. The authors described that correction of HIF1a/HIF2a imbalance improves anemia, but reviewer does not know the mechanism by which SGLT2 affects HIF2a. Please add an explanation and the relevant literature.
àThanks for this comment. In the part of 2.4, we added much more explanation about this part in the end of that paragraph. “The overexpression of HIF-1α was considered as inflammatory effect on renal injury and the activation of HIF-2α can counteract the inflammation and reduce injury in CKD. In a preclinical study, an SGLT2 inhibitor (luseogliflozin) can reduce the protein expression of HIF-1α expression in human renal proximal tubular epithelial cells. Besides, luseogliflozin can also inhibit HIF-1α gene expression, including PAI-1, VEGF, GLUT1, HK2 and PKM. Then luseogliflozin also attenuated cortical tubular HIF-1α expression in db/db mice. Moreover, SGLT2i can upregulated both SIRT1 and AMPK. The increasing SIRT1 and AMPK can further suppress HIF-1α and promote HIF-2α.”
- Protective effects of HIF-PH inhibitors on animal models of diabetic nephropathy have also been reported, suggesting a suppressive effect of HIF1 on MCP1 (e.g. PMID: 31996409). Please cite these references and discuss the significance of HIF and HIF-PH inhibitors in diabetic kidney injury.
àThanks for this comment. We cited this article. (in the part of 2.4: In a preclinical study108, Enarodustat (PHD inhibitor) can activate HIF and then protect against metabolic disorders and associated kidney disease in obese type 2 diabetic mice.)
- There are errors in the description; second line from the bottom on page 6. “oxidative damageiananimal[87].” In figure 2, “more antiplatlet” (upper right) and “Ppoor response to EPO” (lower left).
àThanks for this comment. We revised it accordingly.

Reviewer 2 Report
This review shows new implication for diabetic nephropathy. Although the contents of the manuscript is important, the structure is confusing due to the mixture of drugs and mechanism in each paragraph. Thus, the main message is ambiguous.
Major comments
- The title of this manuscript includes 4 mechanisms (glomerular hyperfiltration, anemia, hypoxia and energy imbalance) linked to diabetic nephropathy, but subheadings in each mechanism make the reviewer difficult to understand.
- The authors may want to focus on HIF stabilizer and SGLT2 inhibitors. If so, two major classification by these drugs, but not mechanisms, are suitable.
- The figures (Figure 1 and 2) are difficult to understand. Figure legends and full names of abbreviations are needed. In addition, figure composition should be drastically changed.
Minor comments
- In the abstract, the full name of “KDIGO” is written after the description of abbreviation.
- In page 2 line 4, “in the United States” should be added.
- In page 2 line 14, “2007” is not “recent”.
- In page 5 line 28, the difference between stage 3 and stage 3A CKD is not clear.
- Ref 124 is not the study about SGLT2 inhibitor, although the authors describe “SGLT2i is thought to balance sodium and calcium homeostasis and rescue mitochondrial function”.
- Ref 135 is not the reference written by Layton, although the authors described Layton wrote.
Author Response
Reviewer 2
Comments and Suggestions for Authors
This review shows new implication for diabetic nephropathy. Although the contents of the manuscript is important, the structure is confusing due to the mixture of drugs and mechanism in each paragraph. Thus, the main message is ambiguous.
Major comments
- The title of this manuscript includes 4 mechanisms (glomerular hyperfiltration, anemia, hypoxia and energy imbalance) linked to diabetic nephropathy, but subheadings in each mechanism make the reviewer difficult to understand.
àSorry about this problem. We revised the title of this article to “ New implication for diabetic nephropathy from bed to bench”. We tried to not highlight these four mechanism in this review article.
- The authors may want to focus on HIF stabilizer and SGLT2 inhibitors. If so, two major classification by these drugs, but not mechanisms, are suitable.
àThanks for this comment. Indeed, they are two classifications of drugs, other than mechanism. We rephrased it in this article. We also added one sentence in the end of abstract to clarify this part “Here, we present a review on the pivotal and new mechanisms of DN from implications of clinical studies and medications. ” We also adjusted the title of 2.4.
2.4 Two hallmark mechanisms of renal anemia and new treatments (HIF stabilizer and SGLT2i) à Two new treatments for renal anemia(HIF stabilizer and SGLT2i)
- The figures (Figure 1 and 2) are difficult to understand. Figure legends and full names of abbreviations are needed. In addition, figure composition should be drastically changed.
àWe added all abbreviation in the figure legends. We also deleted all paragraph mark in the figure.
Minor comments
- In the abstract, the full name of “KDIGO” is written after the description of abbreviation.
àWe revised it accordingly.
- In page 2 line 4, “in the United States” should be added.
à We revised it accordingly.
- In page 2 line 14, “2007” is not “recent”.
àThanks for this comment. We rephrased it as “In the guidelines from National Kidney Foundation in 2007,”
- In page 5 line 28, the difference between stage 3 and stage 3A CKD is not clear.
àWe deleted the data of stage 3A to avoid misunderstanding.
- Ref 124 is not the study about SGLT2 inhibitor, although the authors describe “SGLT2i is thought to balance sodium and calcium homeostasis and rescue mitochondrial function”.
àWe removed this citation. We cited another study (Curr Med Res Opin. 2016 Aug;32(8):1375-85.) regarding SGLT2i on the mineral metabolism.
- Ref 135 is not the reference written by Layton, although the authors described Layton wrote.
àWe deleted “according to Layton”. Thanks for this comment.

Reviewer 3 Report
In the present review, Dr Tsai et al. highlight the impact of recent therapeutic advances on the pathogenic mechanisms of DKD. Some mechanisms are long time known, but some others have been better described due to the recent progress in diabetic nephropathy treatment. I would like to add some comments and suggestions that could help the authors improve the quality of this manuscript:
Major comments:
Comment 1, I think "New approaches to diabetic nephropathy…" instead of "New implication for diabetic nephropathy…" could be a more appropriate title.
Comment 2, Recent KDIGO guidelines recommend avoiding both terms diabetic nephropathy (DN) and diabetic kidney disease (DKD). However, the reviewer thinks there is a difference between concepts that could help the authors when naming the diseases throughout the manuscript. DKD is a diagnosis based on clinical and laboratory findings (PMID 28522654), whereas DN should only be used when a patient has a biopsy confirmed nephropathy. Moreover, a DN diagnosis should be accompanied by Tervaert's classification (PMID 20167701).
Comment 3, In line with the previous comment, I would not say diabetes is the main cause of ESKD. In most cases, the causality is not confirmed. It is more accurate saying "DKD is the leading cause of ESKD" or "Diabetes is the leading acquired risk factor for an accelerated progression of CKD".
Comment 4, Authors could already add that EMPA-KIDNEY trial will show promising results because it has been stopped due to early efficacy:
https://www.empakidney.org/news/empa-kidney-trial-stops-early-due-to-evidence-of-efficacy
Comment 5, I do not know what the authors were trying to explain about SGLT2i in the following sentence "This effect cannot fully be explained by the initial diuretic effect related to hemochromatosis". What does the diuretic effect related to hemochromatosis mean? Did the authors try to say hemoconcentration?
Comment 6, I would recommend authors review the following sentence "Second, there are also many disadvantages regarding cardiac complications and severe infections that result from long-term iron supplementation". I agree that iron supplementation could increase the risk of infection, but I do not think it increases CV death. In fact, higher iron doses could be related to slightly lower mortality (PMID 30365356). It is also true that patients with high ferritin (>500 ng/mL) should receive less IV iron, and if the increase in ferritin is linked to anemia, check why this is happening (infection, subacute inflammatory process…). Iron supplementation is a cornerstone in ESKD, but also in heart failure patients with iron deficiency. Therefore, the sentence can be misleading.
Comment 7, I would say roxadustat is equivalent to darbepoetin alfa, and the superior effects regarding hepcidin decrease are yet to be determined on clinical outcomes. For the moment, no greater beneficial effects have been identified for roxadustat, and recently Kidney International retracted a publication of the drug:
https://pubmed.ncbi.nlm.nih.gov/35257087/
Comment 8, I think authors could explain better the mechanisms through which SGLT2i prevent anemia and maybe add a third figure. It is a fascinating topic that could increase the impact of this manuscript. The reduction of tubular ROS generation and decrease of cortical O2 consumption plays a crucial role and is linked to HIF balance (extend Renal hypoxia and HIF in DN section). Authors should check the following review: PMID 32152499.
Comment 9, In Figure 2, I do not understand what the dashed line between production and loss is trying to explain. I would also avoid subheadings in the Kidney text box.
Minor comments:
Comment 10, In Figure 1 and Figure 2, some words are still underlined as errors detected by Word. Please correct.
Comment 11, Consider including colours in both Figures and adding a short explanation in the Figures' text. In Figure 1, for example, it seems that RASi and ET-1RAs are the cause of efferent arteriolar vasoconstriction.
Comment 12, In the following sentence, "Finally, endothelin-A receptor antagonists from the SONOR study showed reduced risk of renal events in patients with DN", SONOR should be replaced for SONAR.
Comment 13, Consider rewriting the sentence "Renal anemia may occur earlier and worse" to "Renal anemia may develop earlier and be worse in DM related CKD".
Comment 14, review the writing in this sentence "As reported in a cross-sectional study on DM patients, up to 23% of them have anemia as reported al".
Comment 15, In the following sentence, "the renal anemia could contribute the progression of renal function in DM related CKD", I think the authors wanted to say "the renal anemia could contribute to the progression of renal dysfunction in DM related CKD".
Comment 16, Please correct the end of the following sentence "In 2016, in a study on DN animal models, EPO is reported to have suppressed inflammatory response, along with oxidative damageiananimal".
Comment 17, Throughout the text there are several grammatical and spelling errors. I have highlighted some. Please check the text carefully and correct them.
Author Response
Reviewer 3
In the present review, Dr Tsai et al. highlight the impact of recent therapeutic advances on the pathogenic mechanisms of DKD. Some mechanisms are long time known, but some others have been better described due to the recent progress in diabetic nephropathy treatment. I would like to add some comments and suggestions that could help the authors improve the quality of this manuscript:
Major comments:
Comment 1, I think "New approaches to diabetic nephropathy…" instead of "New implication for diabetic nephropathy…" could be a more appropriate title.
à We revised it accordingly.
Comment 2, Recent KDIGO guidelines recommend avoiding both terms diabetic nephropathy (DN) and diabetic kidney disease (DKD). However, the reviewer thinks there is a difference between concepts that could help the authors when naming the diseases throughout the manuscript. DKD is a diagnosis based on clinical and laboratory findings (PMID 28522654), whereas DN should only be used when a patient has a biopsy confirmed nephropathy. Moreover, a DN diagnosis should be accompanied by Tervaert's classification (PMID 20167701).
àThanks for this comment. We cited PMID 28522654 and PMID 20167701 and made some revisions.
Comment 3, In line with the previous comment, I would not say diabetes is the main cause of ESKD. In most cases, the causality is not confirmed. It is more accurate saying "DKD is the leading cause of ESKD" or "Diabetes is the leading acquired risk factor for an accelerated progression of CKD".
àThanks for this comment. We deleted our previous sentence. We used your comment as a new one. “Diabetes is the leading acquired risk factor for an accelerated progression of CKD”
Comment 4, Authors could already add that EMPA-KIDNEY trial will show promising results because it has been stopped due to early efficacy:
https://www.empakidney.org/news/empa-kidney-trial-stops-early-due-to-evidence-of-efficacy
àWe cited this reference in the text. Thanks for this comment.
Comment 5, I do not know what the authors were trying to explain about SGLT2i in the following sentence "This effect cannot fully be explained by the initial diuretic effect related to hemochromatosis". What does the diuretic effect related to hemochromatosis mean? Did the authors try to say hemoconcentration?
àSorry for this typo. We wanted to mean hemoconcentration, instead of hemochromatosis.
Comment 6, I would recommend authors review the following sentence "Second, there are also many disadvantages regarding cardiac complications and severe infections that result from long-term iron supplementation". I agree that iron supplementation could increase the risk of infection, but I do not think it increases CV death. In fact, higher iron doses could be related to slightly lower mortality (PMID 30365356). It is also true that patients with high ferritin (>500 ng/mL) should receive less IV iron, and if the increase in ferritin is linked to anemia, check why this is happening (infection, subacute inflammatory process…). Iron supplementation is a cornerstone in ESKD, but also in heart failure patients with iron deficiency. Therefore, the sentence can be misleading.
àThanks for this comment. We rephrased it as “Second, there are also some disadvantages regarding cardiac complications and infections that result from over iron supplementation”
Comment 7, I would say roxadustat is equivalent to darbepoetin alfa, and the superior effects regarding hepcidin decrease are yet to be determined on clinical outcomes. For the moment, no greater beneficial effects have been identified for roxadustat, and recently Kidney International retracted a publication of the drug:
https://pubmed.ncbi.nlm.nih.gov/35257087/
àThanks for this comment. We added a sentence in the text. “At least, the effect of Roxadustat on renal anemia is like EPO.” We try to not over-emphasize the effect of Roxadustat compared to EPO.
Comment 8, I think authors could explain better the mechanisms through which SGLT2i prevent anemia and maybe add a third figure. It is a fascinating topic that could increase the impact of this manuscript. The reduction of tubular ROS generation and decrease of cortical O2 consumption plays a crucial role and is linked to HIF balance (extend Renal hypoxia and HIF in DN section). Authors should check the following review: PMID 32152499.
àThanks for this comment. We added a new figure 3.
Comment 9, In Figure 2, I do not understand what the dashed line between production and loss is trying to explain. I would also avoid subheadings in the Kidney text box.
àWe re-drew it again. Thanks for this comment.
Minor comments:
Comment 10, In Figure 1 and Figure 2, some words are still underlined as errors detected by Word. Please correct.
àWe re-drew it again. Sorry about that.
Comment 11, Consider including colours in both Figures and adding a short explanation in the Figures' text. In Figure 1, for example, it seems that RASi and ET-1RAs are the cause of efferent arteriolar vasoconstriction.
àThanks for this comment. We add all abbreviations and some figures legends.
Comment 12, In the following sentence, "Finally, endothelin-A receptor antagonists from the SONOR study showed reduced risk of renal events in patients with DN", SONOR should be replaced for SONAR.
àSorry about this typo.
Comment 13, Consider rewriting the sentence "Renal anemia may occur earlier and worse" to "Renal anemia may develop earlier and be worse in DM related CKD".
àThanks for this comment. We rephrased it accordingly.
Comment 14, review the writing in this sentence "As reported in a cross-sectional study on DM patients, up to 23% of them have anemia as reported al".
- Sorry about this typo. It should be “As reported in a cross-sectional study on DM patients, up to 23% of them have anemia as reported by Thomas et a”
Comment 15, In the following sentence, "the renal anemia could contribute the progression of renal function in DM related CKD", I think the authors wanted to say "the renal anemia could contribute to the progression of renal dysfunction in DM related CKD".
àThanks for this comment. We revised it accordingly.
Comment 16, Please correct the end of the following sentence "In 2016, in a study on DN animal models, EPO is reported to have suppressed inflammatory response, along with oxidative damageiananimal".
àSorry about this typo. It should be “In 2016, in a study on DN animal models, EPO is reported to have suppressed inflammatory response, along with oxidative damage in an animal model”.
Comment 17, Throughout the text there are several grammatical and spelling errors. I have highlighted some. Please check the text carefully and correct them.
àSorry about this problem. We review it throughout the text.

Round 2
Reviewer 1 Report
The manuscript was revised in response to reviewer suggestions.
Author Response
Thanks for your comment.

Reviewer 2 Report
The revised manuscript is well modified. One additional modification is necessary.
In Figure 1, the authors should add arrows that mean the suppression by treatment drugs.
Author Response
àThanks for this comment. We revised it accordingly.
